,

# Stratospheric aerosol radiative forcing simulated by the chemistry climate model EMAC using aerosol CCI satellite data

**Christoph Brühl**[1], **Jennifer Schallock**[1], **Klaus Klingmüller**[1], **Charles Robert**[2], **Christine Bingen**[2], **Lieven Clarisse**[3], **Andreas Heckel**[4], **Peter North**[4], **and Landon Rieger**[5]

[1]Max Planck Institute for Chemistry, Mainz, Germany
[2]BIRA Brussels, Belgium
[3]ULB Brussels, Belgium
[4]Swansea University, UK
[5]University of Saskatchewan, Canada

**Correspondence:** Christoph Brühl (christoph.bruehl@mpic.de)

**Abstract.** This paper presents decadal simulations of stratospheric and tropospheric aerosol and its radiative effects by the chemistry general circulation model EMAC constrained with satellite observations in the framework of the ESA-Aerosol-CCI project such as GOMOS (Global Ozone Monitoring by Occultation of Stars) and (A)ATSR ((Advanced) Along Track Scanning Radiometer) on the ENVISAT (European Environmental Satellite), IASI (Infrared Atmospheric Sounding Interferometer) on MetOp (Meteorological Operational Satellite), and, additionally, OSIRIS (Optical Spectrograph and InfraRed Imaging System). In contrast to most other studies, the extinctions and optical depths from the model are compared to the observations at the original wavelengths of the satellite instruments covering the range from UV (ultraviolet) to terrestrial IR (infrared). This avoids conversion artifacts and provides additional constraints for model aerosol and interpretation of the observations.

MIPAS (Michelson Interferometer for Passive Atmospheric Sounding) $SO_2$ limb measurements are used to identify plumes of more than 200 volcanic eruptions. These three-dimensional $SO_2$ plumes are added to the model $SO_2$ at the eruption times. The interannual variability of aerosol extinction in the lower stratosphere, and of stratospheric aerosol radiative forcing at the tropopause is dominated by the volcanoes. To explain the seasonal cycle of the GOMOS and OSIRIS observations, desert dust simulated by a new approach and transported to the lowermost stratosphere by the Asian summer monsoon and tropical convection turns out to be essential. This applies also to the radiative heating by aerosol in the lowermost stratosphere. The existence of wet dust aerosol in the lowermost stratosphere is indicated by the patterns of the wavelength dependence of extinction in observations and simulations. Additional comparison with (A)ATSR total aerosol optical depth at different wavelengths and IASI dust optical depth demonstrates that the model is able to represent stratospheric as well as tropospheric aerosol consistently.

## 1 Introduction

Climate effects of stratospheric aerosols can be important, as analyzed for example by Solomon et al. (2011), Santer et al. (2014) and Ridley et al. (2014). Stratospheric aerosol exerts a negative radiative forcing on the troposphere because enhanced scattering by the particles reduces solar radiation reaching the surface and the lower atmosphere. In addition, changes in diffuse light fraction have shown their potential to enhance photosynthesis (Gu et al., 2003). The aim of the present paper is to use jointly model simulations and satellite observations, taking into account the multiple spectral channels of the instruments to better understand the spatiotemporal evolution of the stratospheric aerosol burden and the contribution of the different aerosol types to the observed dynamical aerosol patterns at the different altitudes. Most earlier studies focus on the effects of major volcanic eruptions like Pinatubo (e.g. Aquila et al., 2012; English et al., 2013). For the post Pinatubo-period with only medium size eruptions Mills et al. (2016, 2017) present simulations with the chemistry climate model WACCM (Whole Atmosphere

Community Model) with interactive aerosol, using estimates for volcanic injections mostly from nadir sounders. That and the present study contribute to the SPARC/SSIRC initiative (Stratosphere-troposphere Processes And their Role in Climate/ Stratospheric Sulfur and Its Role in Climate, see for example Timmreck et al. (2018)), aiming at a better understanding of the composition, microphysical and radiative properties characteristics of stratospheric aerosols and their impact on climate (Kremser et al., 2016). In this work, we rely on the multiple instrument satellite dataset provided in the Climate Change Initiative (CCI) of the European Space Agency (ESA) (Popp et al., 2016), which was developed as tool for evaluation and improvement of the treatment of stratospheric and tropospheric aerosols in global chemistry climate models, like the EMAC (ECHAM5/MESSy Atmospheric Chemistry) model (Brühl et al., 2015). The datasets providing extinctions or total optical depth at wavelengths from ultraviolet (UV) to terrestrial infra-red (IR) are very useful to validate and optimize assumptions on the size distribution and on the composition of aerosol in the model, but also on aerosol sources. Some aspects of the stratospheric part of this study follow up Bingen et al. (2017). The ATSR and IASI datasets provide additional constraints on tropospheric aerosol, especially desert dust. We find in the present study that this latter aerosol compound can penetrate the tropopause via the Asian Summer Monsoon system and, to a smaller extent, via tropical convection.

The present paper is organized as follows: In Section 2, we briefly present the satellite datasets used to evaluate the model, and to check for consistency of observations at different wavelengths: GOMOS, IASI, (A)ATSR and OSIRIS. In Section 3 we describe the EMAC model and the various versions and resolutions used in our work, including the use of MIPAS $SO_2$ for input. In Section 4, we study the impact of the main aerosol sources on the upper tropospheric and lower stratospheric aerosol burden. The influence of volcanic sources derived from satellite data, but also of dust and organic aerosols is analyzed. We present examples on the constraints by satellite observations in different spectral regions on different aerosol types with respect to particle size and composition. We discuss the evolution of the optical depth and radiative forcing by stratospheric aerosols, including uncertainties introduced be horizontal model resolution. Finally, we show that the findings concerning the importance of dust for the lower stratosphere are consistent with observations and simulations of tropospheric aerosol. Conclusions are drawn in Section 5.

## 2 Satellite data products from Aerosol-cci II

### 2.1 GOMOS (Global Ozone Monitoring by Occultation of Stars)

GOMOS is an instrument based on the stellar occultation technique (Bertaux et al., 2010) and provides atmospheric measurements in the UV-Visible-IR range (248-690 nm, 755-774 nm, and 926-954 nm). The use of stellar occultation results in a high rate of occultation measurements, and consequently, a very good spatial coverage compared to solar occultation. As a drawback, the signal-to-noise ratio of each measurement is much lower than in the solar case, and varies with the star characteristics (especially its magnitude and its temperature). The operational retrieval, IPF, provides density profiles for trace gases such as ozone ($O_3$), nitrogen dioxide ($NO_2$), and nitrogen trioxide ($NO_3$) (Kyrola et al., 2010), as well as aerosol extinction. However, the extinction shows a poor quality out of the reference wavelength at 500 nm. For this reason an alternative inverse algorithm called AerGOM was developed specifically to optimize the aerosol retrieval (Vanhellemont et al., 2016; Robert et al., 2016). AerGOM provides vertical profiles of the same gas species, and the total extinction coefficient for the non-gaseous species and its spectral dependence, currently over the range 250-750 nm. The nature of the total extinction fraction for non-gaseous species is then inferred using simple criteria based on the geolocation, associated temperature value and extinction value, and each point of the vertical extinction profile is attributed to aerosols, cirrus clouds, polar stratospheric clouds or meteoritic dust.

From the AerGOM extinction, climate data records (CDRs) were developed in the framework of the ESA Aerosol CCI project for different quantities including the aerosol extinction and the related aerosol optical depth at several wavelengths (355, 440, 470, 550 and 750 nm, Bingen et al., 2017). A particular attention was paid to the grid choice, which should optimally render the information contained in the GOMOS measurement set. The most important conclusions of this optimization were that grid resolution should be chosen to ensure a reasonable statistical sampling in most of the grid cells, and that it should optimally reflect the typical transport of volcanic plumes after an eruption reaching the upper troposphere or the lower stratosphere (UTLS). Therefore, the grid should represent in a coherent way the longitudinal and latitudinal air mass transport, and the time needed for this transport. Also, the temporal resolution should be short enough to enable the detection of volcanic signatures, taking into account the typical lifetime of the plume. In this respect, we could verify that time intervals of about 5 days are able to represent the signature of most of the eruptions injecting sulfuric gases in the UTLS, while such signature is often diluted, underestimated, or even disappears in the case of coarser grid cells. This is the case, for instance, for monthly zonal means, even though this representation is very

commonly used in the field. The ability of the grid to reproduce the signature of volcanic plume in a satisfactory way is of particularly great importance when the CDRs are used to constrain climate models. More detail about the investigations of the optimal grid choice and all other aspects of the implementation of the CDRs can be found in (Bingen et al., 2017).

In their current version (version 3.0), these CDRs are defined on a grid with a resolution of 5° in latitude, 60° in longitude, 1 km in altitude, and 5-day time period. The records cover the whole ENVISAT period (March 2002 - April 2012) and include the total extinction of non-gaseous species, but also the polar stratospheric cloud (PSC) fraction and the cloud-free aerosol fraction which is dominated by sulfate aerosols below an altitude of 32 km. It is important to mention that cloud detection is not yet optimal, and that cloud contamination of the aerosol fraction is possible in the UTLS region. This issue is still under investigation.

## 2.2    IASI (Infrared Atmospheric Sounding Interferometer)

The IASI dust dataset of the Université Libre de Bruxelles (ULB) was generated in the context of ESA CCI's project (Popp et al., 2016). It is based on a statistical regression technique and the use of a neural network trained on synthetic IASI data. A similar scheme has already been applied for the retrieval of $NH_3$ (ammonia) (Whitburn et al., 2016; Van Damme et al., 2017). As input variables it uses the IASI L2 pressure, humidity and temperature information, spectral information and a CALIPSO (Cloud-Aerosol Lidar and Infrared Pathfinder Satellite Observation) derived dust altitude climatology. The main output variables are dust optical depth at 10 and 11 $\mu$m (and 550 nm). Initial results and validation performance are provided in (Popp et al., 2016).

## 2.3    (A)ATSR ((Advanced) Along Track Scanning Radiometer)

The ATSR (SU) algorithm has been developed at Swansea University for estimation of atmospheric aerosol and surface reflectance for the ATSR-2, AATSR sensors, and SLSTR (Sea and Land Surface Temperature Radiometer) on Sentinel-3. Over land, the algorithm employs a parameterized model of the surface angular anisotropy (North, 2002), and uses the dual-view capability of the instrument to allow aerosol property estimation without a priori assumptions on surface spectral reflectance. Over ocean, the algorithm uses a simple a priori model of ocean surface reflectance at both nadir and along-track view angles. A climatology (Kinne et al., 2006) is used to constrain chemical composition of the aerosol components at 1° x 1° latitude-longitude grid, while the method retrieves aerosol size and optical thickness on a 10 km grid. Both optical thickness and size are retrieved as vertical column values. Size is not resolved vertically, but is

represented by fraction of fine and coarse mode aerosol in the total. The algorithm has been developed from initial prototype (Bevan et al., 2012) under the Aerosol CCI program, and results and validation performance for version 4.21 are provided in Popp et al. (2016). The version used here (V4.3) differs from that summarized in Popp et al. (2016) by improvements in retrieval of coarse/fine mode fraction, and improved cloud screening over ocean in the region of dense plumes, resulting in approximately 10% greater coverage, with small improvement in correlation against AERONET (AErosol RObotic NETwork) values. AERONET is recognized as reference dataset for validation of satellite data products (Holben et al., 1998).

## 2.4    OSIRIS (Optical Spectrograph and InfraRed Imager), external

OSIRIS was launched on board the Odin satellite, and has provided vertical profiles of limb scattered radiance between 280 and 810 nm since 2001 (Llewellyn et al., 2004). The radiance profiles are inverted to provide aerosol extinction measurements at 750 nm at altitudes between 10 and 35 km with a vertical resolution of approximately 2 km (Bourassa et al., 2012). This technique provides high sampling rates with hundreds of measurements per day over the sunlit portion of the globe, enabling excellent spatial and temporal sampling of short-lived events. OSIRIS aerosol extinction retrievals agree well with coincident occultation measurements from Stratospheric Aerosol and Gas Experiments II and III during background periods but have known low biases above approximately 25 km, and will have some cloud contamination near and below the tropopause (Bourassa et al., 2012; Rieger et al., 2015). Additionally, seasonal biases are possible due to the orbital geometry and changes in aerosol optical properties such as after volcanic eruptions may also bias the retrievals. These effects are described in more detail by Rieger et al. (2014, 2018). This work uses the OSIRIS version 5.10 aerosol retrieval (Bourassa et al., 2018) averaged into daily, 5° latitude by 30° longitude bins for comparisons.

## 3    Model Setup

For the simulations of the radiative and chemical effects of stratospheric aerosol, the ECHAM5 (5th generation of European Centre Hamburg general circulation model) general circulation model coupled to the Modular Earth Submodel System Atmospheric Chemistry (EMAC) was used (Brühl et al. (2015), updated to the version of Jöckel et al. (2010). In contrast to Jöckel et al. (2016), who use stratospheric aerosol extinction climatologies derived from observations, in our model setup aerosol and its optical properties are calculated from precursor gases and emissions. As dust reaching the upper troposphere/lower stratosphere region (UTLS) turned out to be sensitive to model resolution, we used different model

resolutions: the T42 resolution (spectral, 2.75° in latitude and longitude) of the previous studies, T63 resolution (1.88°), the standard resolution for the stratosphere used in this study and T106 resolution (1.1°) for a one year sensitivity test. The vertical grid has 90 layers from the surface up to 0.01 hPa ( 80 km altitude, short L90) with finest resolution in the boundary layer and near the tropopause. For T106 only simulations with the low top model version with 31 levels up to 30 km altitude (L31), the setup used by Klingmüller et al. (2018), which is well tested regarding the representation of tropospheric aerosol, are discussed here in detail. In all simulations, except the T42L90 one of the previous studies, the meteorology below about the 100 hPa level is nudged to the reanalysis ERA-Interim (Jöckel et al., 2006). The simulations were performed for the ENVISAT time period from July 2002 to March 2012 to allow for the use of data from MIPAS for input, and GOMOS and ATSR for validation. The period from 1997 to 2002 using SAGE II (Stratospheric Aerosol and Gas Experiment) was simulated first to get consistent initial conditions.

The applied aerosol module GMXE (Pringle et al., 2010) accounts for seven modes using lognormal size distributions (nucleation mode, soluble and insoluble Aitken, accumulation and coarse modes). The boundary between accumulation mode and coarse mode, a model parameter, is set at a dry particle radius of 1.6 $\mu$m to avoid too fast sedimentation of a too large coarse mode fraction in case of major volcanic eruptions. For dust sensitivity studies in T106 which focus on the troposphere, also a boundary of 1.0 $\mu$m is used. The mode parameters are used for every aerosol type and listed for convenience in Table S1 of the supplement. Optical properties for the types sulfate, dust, organic and black carbon (OC and BC), sea salt, and aerosol water are calculated using Mie-theory-based lookup tables consistent with the selected size distribution widths of the modes. The resulting optical depths, single scattering albedos and asymmetry-factors are used in radiative transfer calculations which (except for the T106 low top sensitivity studies) feedback to atmospheric dynamics. The contribution of stratospheric aerosol to (instantaneous) radiative forcing and heating is calculated online via multiple calls of the radiation module.

The mineral dust emissions are calculated online using the emission scheme of Astitha et al. (2012) which builds on previous studies by Pérez et al. (2006), Spyrou et al. (2010), Laurent et al. (2008, 2010), Marticorena et al. (1997), Zender et al. (2003) and Tegen et al. (2002). The emission scheme parameterizes saltation bombardment and aggregate disintegration by sand blasting, combining the surface friction velocity with descriptions of land cover type, clay fraction of the soil and vegetation cover. For an improved representation of dust at higher resolution, we adopted the updates presented by Klingmüller et al. (2018) in the T106L31 simulation.

Aerosol module parameters, like for example, the composition of sea salt, were optimized on the basis of the satellite data. We apply the chemical speciation of the sea salt emission flux used by Abdelkader et al. (2015) as listed in Table S2 of the supplement. The sea salt composition affects the hygroscopic growth and thereby the AOD. The setting of Jöckel et al. (2016), dominated by Na and Cl ions, which we initially applied in our simulations produced very high AOD levels over the North Pacific which are not consistent with the satellite observations.

$SO_2$ plumes (sulfur dioxide) from about 230 explosive volcanic eruptions into the stratosphere were derived from 3-dimensional data fields of MIPAS (Höpfner et al., 2015) and, in case of data gaps, of GOMOS on ENVISAT with a temporal resolution of 5 days, and added as volume mixing ratio to the simulated $SO_2$ at the time of the eruption. Each identified volcanic eruption (with names from the Smithsonian volcanic database, www.volcano.si.edu) is listed in an emission inventory published recently (Bingen et al., 2017), which provides an estimate of the altitude and the amount of $SO_2$ injected into the atmosphere. The table and the 3-D-fields of volcanic $SO_2$ are available at https://doi.org/10.1594/WDCC/SSIRC_1. These data were derived from MIPAS within the uncertainty range but more near the upper end for best results with the model resolution T42L90 and free running mode, which has some artifacts from the convection scheme and a dry bias at the tropical tropopause. For the nudged T63L90 simulation the volcanic $SO_2$ data of the inventory have to be downscaled by about a factor of 0.7 which is actually closer to the most likely MIPAS measurements. The actual values for each injection, which depend on the time span between the eruptions and on corrections for data gaps, are given in the supplement (Table S3). Boundary conditions for background concentrations of $SO_2$ from outgassing volcanoes into the troposphere are taken from the monthly climatology of Diehl et al. (2012) truncated at 200 hPa to avoid double counting in the stratosphere. The sulfur source gas OCS (carbonyl sulfide) is constrained by observed monthly zonal average surface volume mixing ratios (update of the data by Montzka et al., 2007). Marine DMS (dimethyl sulfide) as natural sulfur source is also included in the model, using a module for exchange fluxes between seawater and atmosphere by Pozzer et al. (2006) and the Lana et al. (2011) climatology. For anthropogenic emissions of CO (carbon monoxide), NOx (nitrogen oxides), sulfur, OC, and BC the DLR-MACCity emission inventory is used. Biomass burning is based on ACCMIP-MACCity and GFEDv2, OC-SOA (secondary organic aerosol) on AEROCOM_UMZ1. For details on these emission inventories selected for the Chemistry Climate Model Initiative (CCMI) see Jöckel et al. (2016).

## 4    Stratospheric Aerosol and its radiative effect

### 4.1    Volcanic eruptions

Volcanic emissions have a large impact on the stratospheric aerosol burden. Even small and moderate eruptions con-
5 tribute to the stratospheric aerosol load due to convective transport of $SO_2$ and its gradual uplift to the upper tropo-sphere and the lower stratosphere, and resulting accumula-tion of sulfate aerosol. Volcanic $SO_2$ injections explain most of the interannual variability of stratospheric aerosol extinc-
10 tion (decadal logarithm) observed by GOMOS, as depicted in Fig. 1 at three wavelengths. For each wavelength (350 nm on Fig. 1ab, 550 nm on Fig. 1cd, and 750 nm on Fig. 1ef, respectively), the GOMOS time series (1ace) showing the al-titude dependence in the tropics, is compared with the EMAC
simulation in resolution T63L90 including the dust contribu-tion (1bdf; see Section 4.2 for more detail). 1 shows, at all three wavelengths, an enhancement of the extinction value is observed around 16-18 km, corresponding to the aerosol load resulting from a succession of volcanic eruptions dur-
ing the whole period 2002-2012. The eruptions of Nabro in June 2011 and the successive eruptions of Soufriere Hills and Rabaul in 2006 have the largest effects on extinction in the lower stratosphere in the observations and the simulation. The best agreement between GOMOS and EMAC is found
in the case of the extinction at 550 nm (Fig. 1cd), where the quality of the GOMOS retrieval is the best. At 750 nm (Fig. 1ef) also, GOMOS measurements agree well with EMAC for the aerosol layer (16-22 km) where measured extinction val-ues exceed $\approx 2 \ 10^{-4} \ km^{-1}$. At lower altitudes (14-16 km),
rather unstructured patterns of enhanced extinction are found by GOMOS, probably corresponding to cloud contamina-tion. At 350 nm, where a decrease of the GOMOS quality is expected due to a loss in signal-to-noise ratio obtained in the UV spectral region while using cold stars, still the vol-
canic events stick out. More details over these aspects can be found in references (Robert et al., 2016; Bingen et al., 2017). Bingen et al. (2017) present also the latitude dependence of 550 nm-aerosol extinction at 17 km altitude as observed by GOMOS and simulated by EMAC in the coarse resolution
T42L90 in their figure 10.

### 4.2    Dust and organics from the troposphere in the upper troposphere/lower stratosphere (UTLS)

Extinction in the lowermost stratosphere and upper tropo-sphere is to a large fraction due to desert dust and organic
carbon aerosol. These contributions were strongly underesti-mated in Brühl et al. (2015) due to a crude parameterization in the used model version based on Jöckel et al. (2006), but overestimated in Bingen et al. (2017). Both simulations were performed in the relatively coarse resolution T42L90. Dust
reaching the UTLS is sensitive to model resolution, mostly via the convection parameterization (Tiedtke, 1989). In Fig.

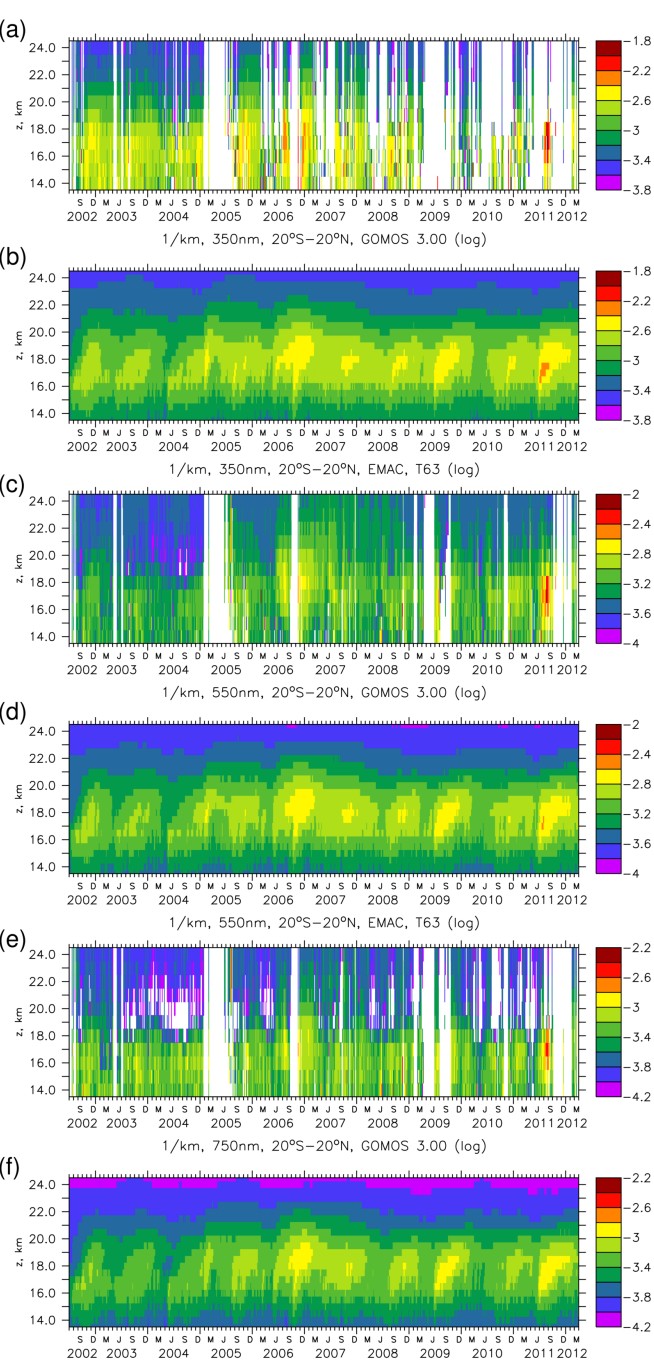

**Figure 1.** GOMOS and EMAC extinctions (log) in the tropics as function of altitude for different wavelengths: (a),(b) UV 350 nm, (c),(d) visible 550 nm and (e),(f) near infrared 750 nm; resolution T63L90.

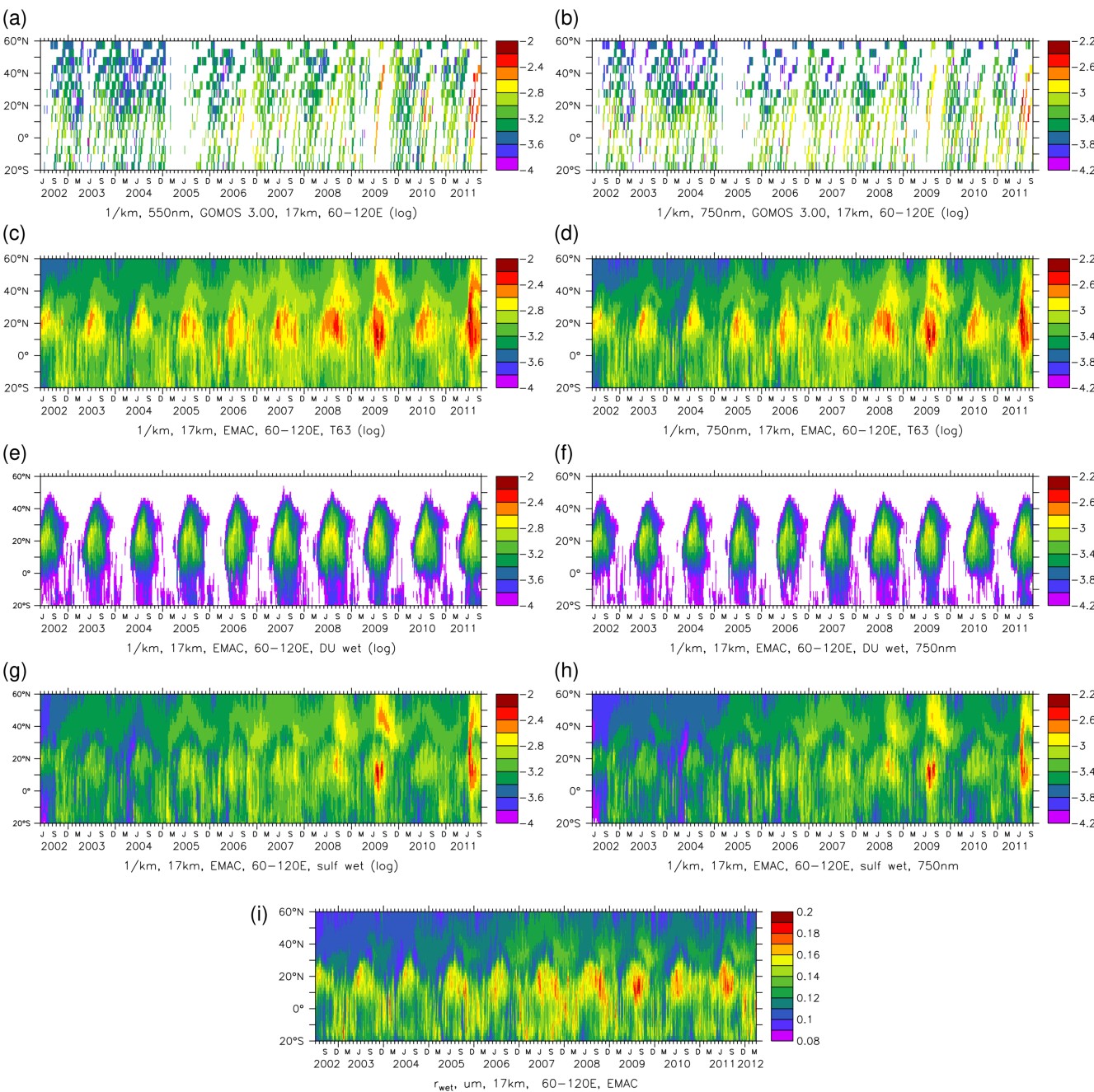

**Figure 2.** Observed ((a),(b)) and simulated ((c),(d), EMAC T63L90) extinction in the Asian sector (60°E-120°E, 20°S- 60°N) for 550 nm ((a),(c)) and 750 nm ((b),(d)). Contribution of wet dust ((e),(f)) and wet sulfate ((g),(h)) to extinction for 550 nm ((e),(g)) and 750 nm ((f),(h)). (i) median wet radius in accumulation mode (for effective radius multiply by 1.4).

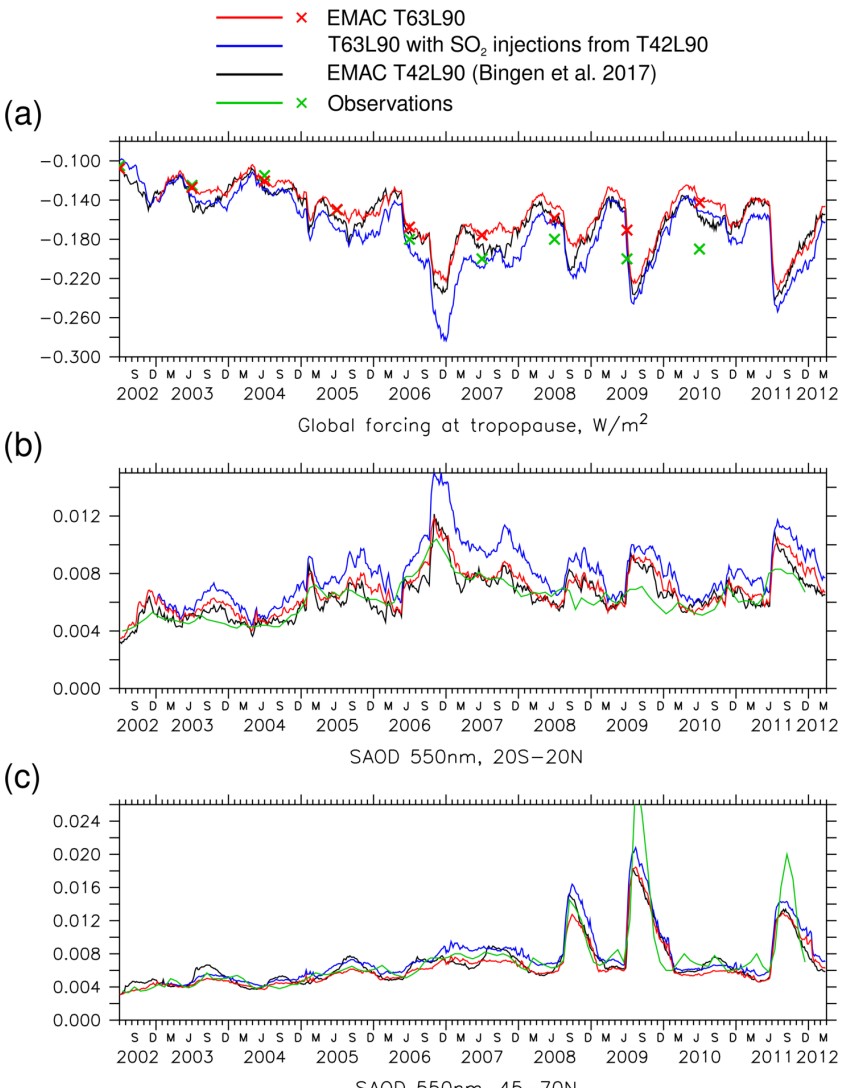

**Figure 3.** (a) Stratospheric aerosol radiative forcing, (b) and (c) stratospheric AOD for tropics and midlatitudes. Red lines and crosses: EMAC, resolution T63L90, current version; black: EMAC T42L90 (Bingen et al., 2017); blue: T63L90 without downscaling the SO₂ injections for T42L90. green: From observations (crosses annual mean for forcing (Solomon et al., 2011), SAGE II, CALIPSO, OSIRIS).

1 the simulated extinction in resolution T63L90 fits well to the GOMOS observations which appear to have a seasonal contribution from the Asian summer monsoon. For more detailed analysis Fig. 2 shows observed and simulated extinc-
5 tion in the Asian sector at 17 km in the visible and the near IR. The largest extinction values are found indeed at the location and time of the Asian summer monsoon at the altitude of outflow. This feature is clearest in years not perturbed by medium strength volcanic eruptions, like for example 2010.
For a clear separation the contributions of wet dust and wet sulfate to extinction are displayed separately (Fig. 2e-h). The wet dust particles in the monsoon region have a larger median wet radius than the volcanic sulfate particles (e.g. from Sarychev in 2009, Fig. 2i) which is consistent with a rela-
tively larger extinction in the infrared compared to the visible

in the monsoon region in observations and simulations. Figure 2a-d demonstrates that dust is essential to reproduce the observations. Total extinction without wet dust in T63L90 is shown in the supplement. Comparing Fig. S1b with Fig. 2g shows a small contribution of organics from biomass burn- 20 ing in northern spring (for volume mixing ratios see Fig. S2). Fig. S1 contains also results from the T42L90 simulation of Bingen et al. (2017), showing that there the contribution of wet dust to extinction has to be downscaled (i.e. devided) by a factor of 2 to get agreement (Fig. S1d, factor of 3 if only 25 dry dust is considered).

Observations by IASI and ATSR indicate a maximum in dust aerosol optical depth (DAOD) in early northern hemispheric summer over the Asian deserts located in the inflow regions of the monsoon (see section 4.4). A similar feature is 30

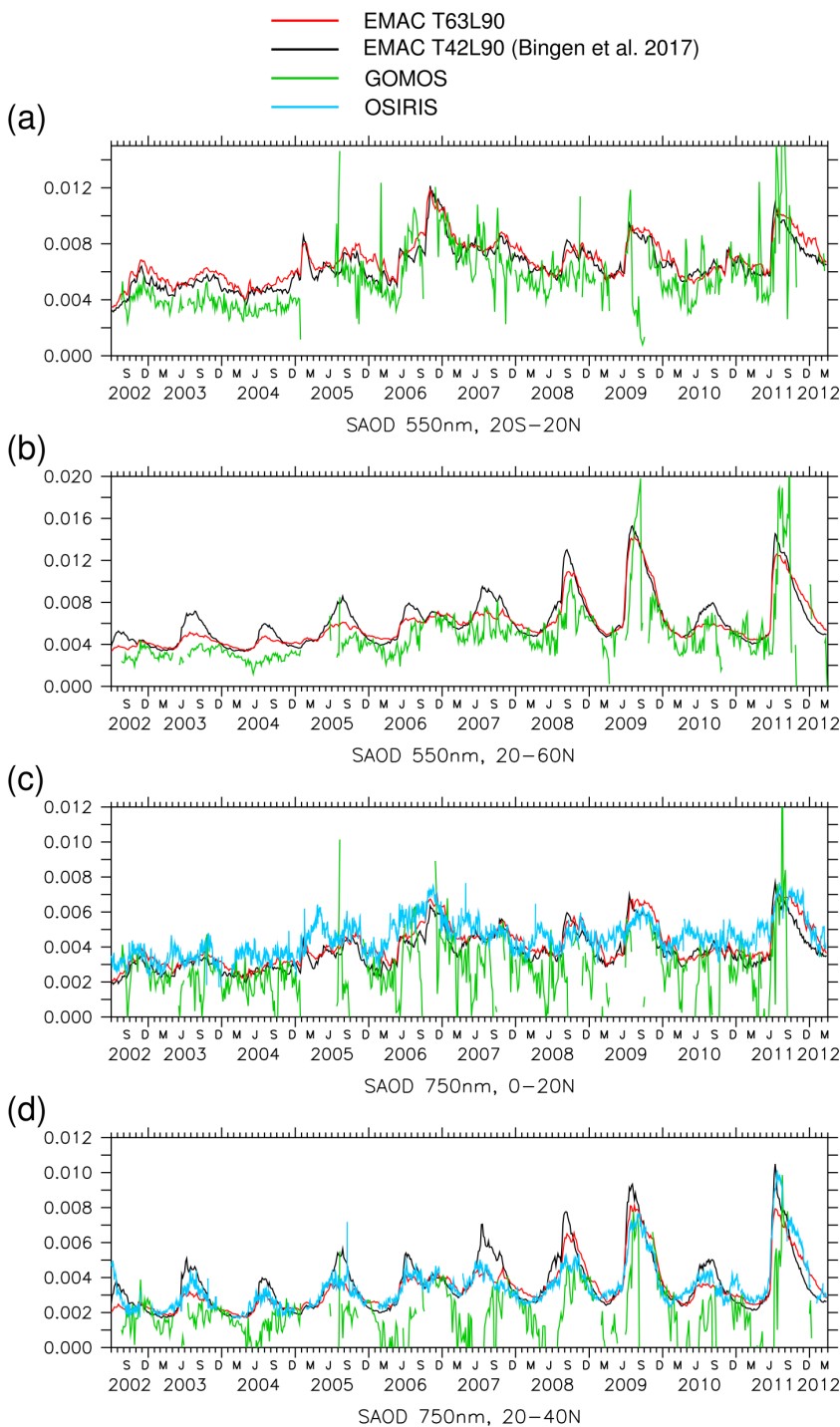

**Figure 4.** (a),(b) Stratospheric AOD at 550 nm observed by GOMOS (green) and simulated by EMAC in resolutions T42L90 (black) and T63L90 (red). (c),(d) Stratospheric AOD at 750 nm in northern tropics and subtropics (SAOD above 15 km), additionally with OSIRIS observations (light blue).

found in the simulations by EMAC. This supports our findings that desert dust is also important for the UTLS.

### 4.3    Stratospheric aerosol radiative forcing, stratospheric aerosol optical depth and radiative heating

Desert dust transported to the UTLS mostly via the Asian summer monsoon contributes significantly to the seasonal cycle of total stratospheric aerosol optical depth (SAOD) in satellite observations and the EMAC simulations shown in Fig. 3b for the tropics (vertical integral of extinction above about 16 km) and in 3c for midlatitudes (above about 14 km). Global radiative forcing at the tropopause is depicted in the 3a. The figure contains in black results from the T42L90 simulation of Bingen et al. (2017) and in blue the T63L90 simulation with the high volcanic sulfur input derived for the coarse resolution. Green lines and symbols show estimates derived from satellite observations (SAGE II, OSIRIS and CALIPSO, Solomon et al., 2011; Santer et al., 2014; Bourassa et al., 2012; Glantz et al., 2014). Red shows results of the current model version in T63L90 with the Astitha et al. (2012) dust scheme and corrected $SO_2$ input (see section 3 and supplement). Concerning global radiative forcing, the volcanoes are the dominating effect with up to 0.13 W/m$^2$ for Rabaul and Nabro compared to the volcanicly quiet period in 2002. Here the use of the $SO_2$ inventory for T42L90 in the T63L90 simulation (blue) causes an overestimate of up to 50% in 2006 and 2007 due to accumulation effects of eruptions following in short sequence. This is visible in the overestimate of tropical SAOD depicted by the blue curve in Fig. 3b.

Especially in Northern midlatitude summer SAOD in T42L90 appears to be high because in that resolution the convective transport of dust to the UTLS in the Asian monsoon region is overestimated (Fig. 3c). This is clearly seen in Fig. 4 which shows in black the T42L90 simulation, in green the observations of 550 and 750 nm SAOD by GOMOS and in light blue (Fig. 4cd only) by OSIRIS in different latitude bands, including the monsoon region. For the narrow latitude bands in Fig. 4cd inclusion of OSIRIS data is important because here GOMOS has often too low coverage. Nevertheless, for a lot of features the two satellite datasets agree well. Using the higher resolution T63L90, for which the convection parameterization was developed, the agreement to the satellite observations is much better (Figs. 3 and 4, red) than with T42L90, especially at midlatitudes and in subtropics. In the subtropics (Fig. 4d) the simulation with low resolution (black) always overestimates the monsoon peaks in August compared to the ones seen in the observations. Comparing the model results with OSIRIS in the northern tropics (Fig. 4c) indicates that some volcanic events are still missing in the inventory, for example in spring 2007 and 2010. This would also explain the differences in radiative forcing (indicated by crosses in Fig. 3a) in these years.

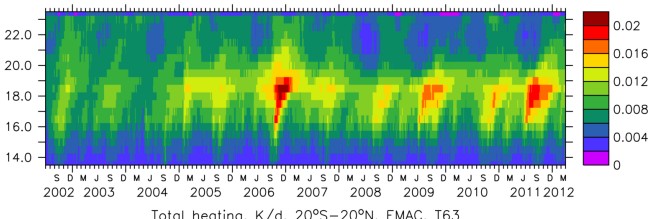

**Figure 5.** Simulated aerosol radiative heating in tropics (solar + infrared, T63L90).

The simulated aerosol radiative heating, derived from radiation calls with and without aerosol, reflects the medium volcanic eruptions with the largest effects near 18 km (Fig. 5). There, desert dust causes additional heating at the time of the Asian summer monsoon. In the UTLS below every year in September a clear signal from biomass burning organic aerosol, its volume mixing ratio is shown in Fig. S2 of the supplement, is visible. Above, around 22 km, the dust below in northern hemispheric summer causes a reduction of absorption of terrestrial radiation by ozone.

### 4.4    Constraints from total aerosol optical depth in different spectral regions and for different aerosol subsets

The first comparisons are carried out for EMAC in T63L90, the standard resolution used in the previous sections. Here AOD refers to troposphere and stratosphere. The DAOD (dust AOD) in terrestrial infrared is most sensitive to the coarse mode of tropospheric dust. Figure 6ab shows that the model reproduces most of the IASI features. DAOD in the visible spectral region (Fig. 6cd) is too high over central Asia, pointing to an overestimate of dust in the accumulation mode near the Taklamakan desert. The patterns in the IR and visible spectral range are different despite considering the factor 2 often applied by AEROCOM/AEROSAT (Aerosol Comparison between Observations and Models) community for conversion in the color scales of Fig 6ab and Fig. 6cd. This holds for model and observations. The fine mode AOD fraction, which is dominated by the accumulation mode, is slightly overestimated over Europe and underestimated in the biomass burning regions in Africa (Fig. 6ef). In the model this is sensitive to the way how the extinction of aerosol water is attributed to the soluble aerosol species, especially sea salt. Absorbing AOD, i.e. AOD times (1-$\omega$) with $\omega$ single scattering albedo, agrees surprisingly well (Fig. 6gh). In the total AOD (Fig. 6ij) there appears to be too much sea salt in the model, or still not optimum parameters for the sea salt composition which controls water uptake (see section 3).

Figure 7 compares the annual average for 2011 of the 10 $\mu$m DAOD observed by IASI and simulated by EMAC in the low top version with high horizontal resolution (T106L31, about 1.1°). The satellite retrievals are taken from version

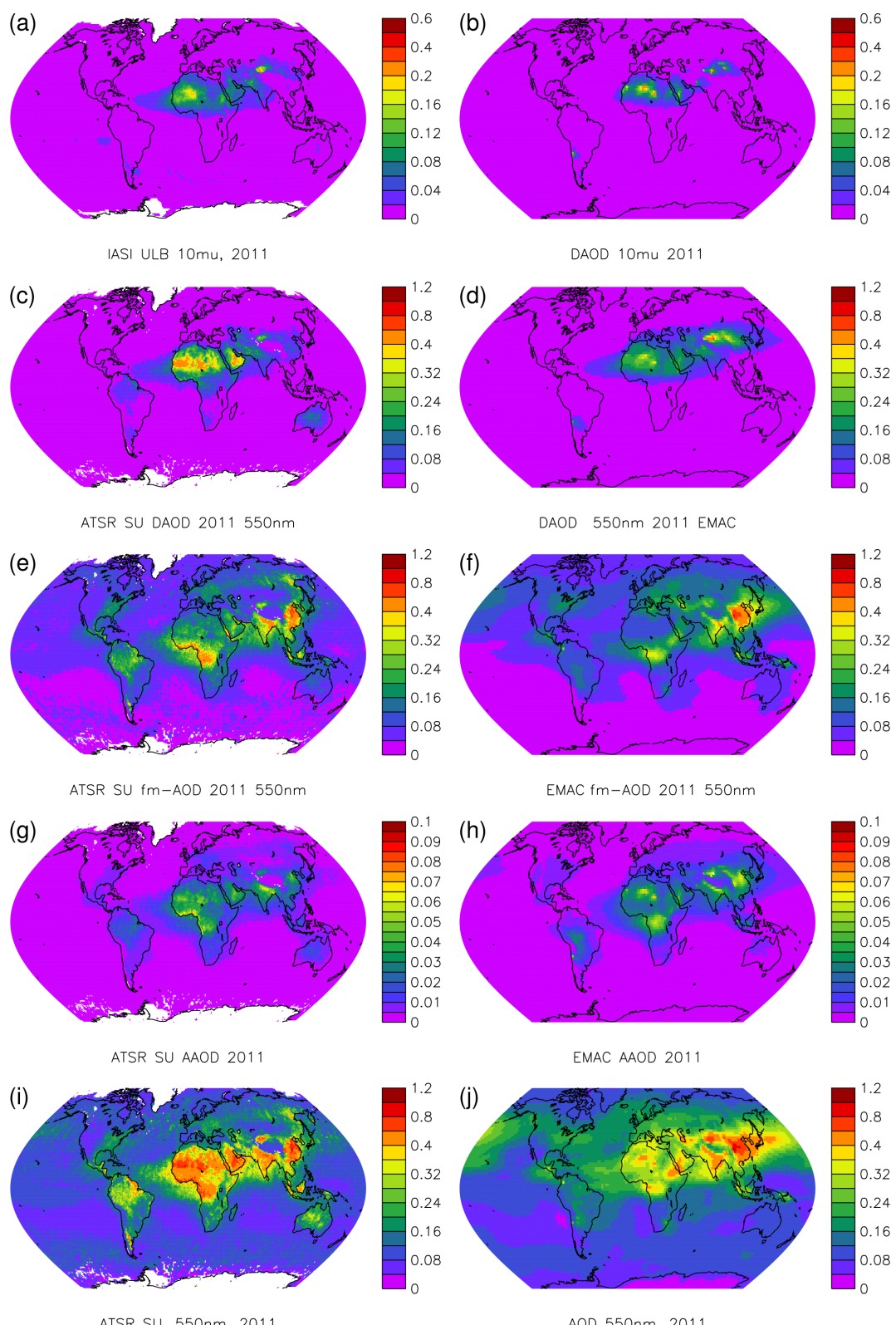

**Figure 6.** Observed (left) and simulated (right) (a),(b): 10 $\mu$m dust AOD (DAOD) for IASI and EMAC; (c),(d): 0.55 $\mu$m DAOD from ATSR and EMAC; (e),(f): fine mode AOD; (g),(h): absorbing AOD (AAOD) and (i),(j): total AOD for ATSR (SU) and EMAC in T63L90 resolution, annual mean 2011.

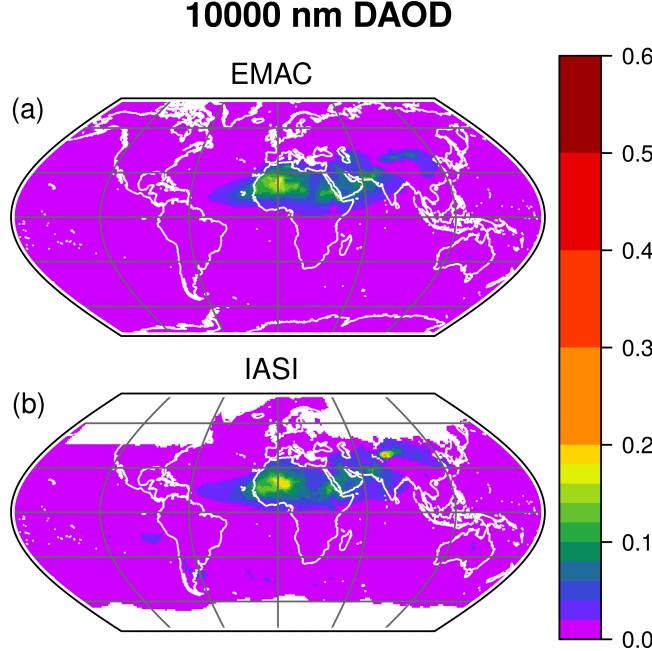

**Figure 7.** Annual mean for 2011 of the DAOD at 10 $\mu$m wavelength observed by IASI ((b), IASI ULB dataset version 8) and simulated by EMAC (a) at T106L31 resolution.

8 of the ULB dataset. The simulation utilizes the dust emission scheme of Klingmüller et al. (2018) which calculates the emissions online considering the meteorological conditions. To extract the DAOD from the total EMAC AOD at 10 $\mu$m,

we apply a filter nullifying sea salt dominated AOD values. To identify the latter, we compare the AOD weighted with the volume of sea salt and dust.

The observed and modelled global DAOD distributions shown in Fig. 7 agree remarkably well. The pixel values of

10 each map are strongly correlated with a correlation coefficient of 0.91. The overall AOD level is consistent as well, so that a similar variance of the pixel values is obtained for the observed (0.00038) and the modelled (0.00041) DAOD distribution. Interestingly, the DAOD from the older version 7

of the ULB dataset yields a pixel by pixel correlation coefficient of only 0.89 and a pixel value variance of only 0.00029. We conclude that the agreement of EMAC and IASI has improved with the update from version 7 to version 8 of the IASI ULB dataset.

The main disagreement of the two maps in Fig. 7 is the less pronounced maximum over the Taklamakan Desert in Central Asia in the model result. This underestimation is related to the model surface friction velocity in mountainous regions like the surroundings of the Taklamakan Desert, which tends

to be lower in simulations at higher horizontal resolution (e.g. T106) than at lower resolution (e.g. T63), possibly resulting in an underestimation of the dust emissions.

Figure 8 compares results from the T106L31 EMAC simulation for the annual average of the total AOD at visible

and near-infrared wavelengths with AASTR retrievals using the ATSR (SU) algorithm version 4.3. Generally good agreement is obtained at 550 nm which is consistent with the good agreement between the 550 nm MODIS (Moderate-resolution Imaging Spectroradiometer) AOD and model re-

sults based on the same EMAC version (Klingmüller et al., 2018). As for the T63L90 simulation, the model yields higher sea salt related AOD levels over the oceans. In contrast, the model AOD over the Sahara is lower than the satellite retrieved values. This becomes even more evident at larger

wavelengths (Fig. 8cf): the model AOD over the Sahara, in contrast to most other regions, has a stronger wavelength dependence than the observed AOD, corresponding to a larger Ångström exponent. This discrepancy might be resolved by adjusting the dust particle size distribution in the model un-

der the constraint of not sacrificing the good agreement of model and observed AOD at 550 nm and at 10 $\mu$m. This could involve modifying the parameters of the log-normal modes, i.e., their widths and boundaries, but also reassessing the parameterization of relevant processes such as emission,

deposition, coagulation and hygroscopic growth, or even adding an extra mode for extremely coarse particles which can be relevant close to dust sources. Over South America, the biomass burning regions of Africa, India and China the wavelength dependence of model and observed AOD is

largely consistent.

## 5   Conclusions

Satellite data are not only important to constrain model parameters; they are very important for model improvement. Comparing satellite data with model results at different

wavelengths simultaneously provides additional information and is also valuable for the satellite community to check internal consistency, as in our case for GOMOS and OSIRIS.

Sophisticated modelling of dust and organic aerosol as well as a detailed volcano dataset are necessary to reproduce

the seasonal cycle and the interannual variability of extinction in the lowermost stratosphere observed by GOMOS at different wavelengths. From the wavelength dependence in observations and simulations regions in the UTLS with enhanced particle size due to water uptake can be identified as

aged dust in the Asian monsoon region. Convective transport of dust into the UTLS is resolution dependent because of differences in convection top height and overshooting convection. A resolution of T63L90 (1.88° in longitude and latitude, 90 vertical layers) fits best to the observations. For the

low resolution T42L90 (2.75°) dust SAOD (and stratospheric mixing ratio) has to be downscaled by a factor of about 0.33, while for higher resolutions (e.g. T106L90) upscaling is required. The resolution dependent differences in convection modify also the residence time of sulfur species in the low-

ermost stratosphere, especially at low latitudes, in resolution T42L90 it appears to be too short.

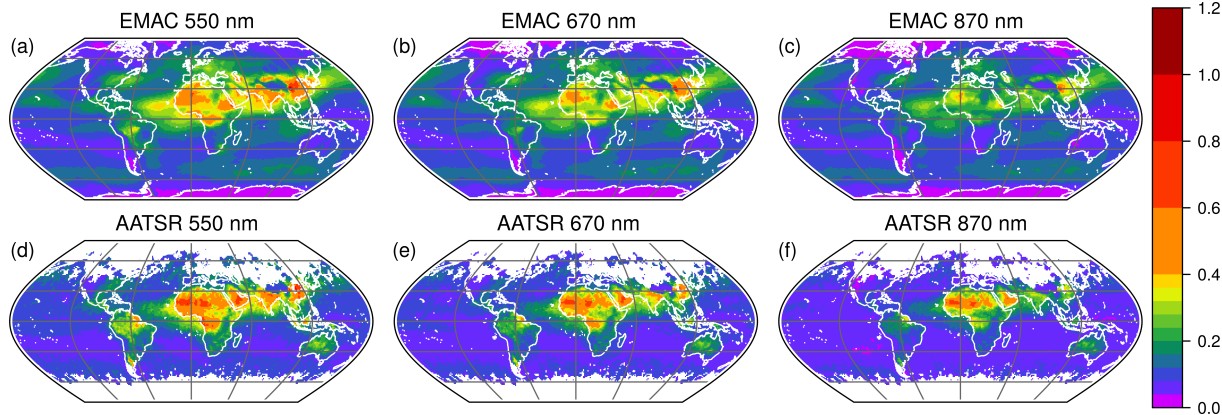

**Figure 8.** Annual mean for 2011 of the AOD at (from left to right) 550 nm, 670 nm and 870 nm wavelength observed by AATSR ((d),(e),(f) SU-ATSR algorithm version 4.3) and simulated by EMAC ((a),(b),(c)) at T106L31 resolution.

The total AOD in the visible spectral range is very sensitive to aerosol water and the composition of sea salt. In the modal model, the bulk fraction has to be increased compared to ions to reduce artifacts of too much water uptake by sea salt. The satellite data helped to identify a preferred parameter set for the sea salt emission composition.

Our simulated dust total aerosol optical depth agrees with satellite data in the visible (ATSR SU) and the infrared (IASI ULB, version 8). The combined comparison at visible and infrared wavelengths provides strong constraints on the modelled particle size distribution. The direct comparison of observations and model reveals different structures in the extinction patterns at both spectral ranges. From this, we conclude that simply assuming a spatially constant factor of (about) 2 for conversion of DAOD from 10 $\mu$m to 550 nm, as commonly applied in the AEROCOM/AEROSAT community, is too crude.

Satellite datasets identifying volcanic $SO_2$, including its vertical distribution or enhanced extinction by aged dust enable the model to get closer to observationally based estimates for radiative forcing, showing the interest of a close interaction between modelling and observation research teams.

*Data availability.* The Aerosol-CCI satellite data are available at ICARE, Lille. All model output of EMAC used here is stored at DKRZ, Hamburg and available on request. This includes 5day averages and 10 hourly values.

*Author contributions.* CB wrote the paper and performed the stratospheric simulations, supported by JS. KK performed the tropospheric simulations and provided code for the stratospheric part, CBi and CR provided the GOMOS data and the corresponding text, LC the IASI data, PN and AH the ATSR data, and LR the OSIRIS data

*Competing interests.* The authors declare that they have no conflict of interest

*Acknowledgements.* This study was funded by the Aerosol CCI project, phase II, of the ESA Climate Change Initiative, as a user option, and by the EU-FP7 project STRATOCLIM. Supporting work for the development of GOMOS datasets was performed in the framework of a Marie Curie Career Integration Grant within the 7th European Community Framework Programme under grant agreement no. 293560. The satellite data, except OSIRIS, were provided via the Aerosol-CCI-database at ICARE, Lille, France; the model simulations were performed at DKRZ, Hamburg, Germany, where also the results are stored.

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
