# Peer review of "Stratospheric aerosol radiative forcing simulated by the chemistry climate model EMAC using aerosol CCI satellite data"

_Atmospheric Chemistry and Physics, 2018_

## Referee Comment (RC1) · Anonymous Referee #2 · 17 May 2018

This paper compares stratospheric aerosol as simulated by different configurations of the global chemistry-climate model EMAC with satellite data from the Climate Change Initiative.

Although it is potentially interesting for the community, the manuscript has several issues mostly related to the quality of the presentation. The writing needs to be completely revised. Detailed comments and suggestions are given below.

GENERAL COMMENTS:

- The introduction is too short and does not put this study in the context of existing literature. Similar papers on the subject shall be cited and related to the work presented

in the manuscript.

- It is also not immediately clear what are the novelty aspect of this study and what is its added value to the current knowledge in the field. It should be stated that the paper focuses on the evaluation of the EMAC model in different configurations, specifically on the aspects controlling the optical and radiative properties of stratospheric aerosol.

- The description of model setup is also not very exhaustive: it is not clear for example how the SO2 plumes from the data are used in the model (last paragraph of Sect. 3). It is also not mentioned which time period is covered by the simulations (although this can be found out later in the figures) and whether a nudging technique has been used.

- As far as I could understand, the setup has been derived from one of the simulations in Jöckel et al. (2016). If this is the case, I would recommend to write that more explicitly at the beginning of Sect. 3. The rest of the section could then discuss just the differences and the additional features considered for the present study. The choice of particular configuration settings shall also be motivated in view of the analysis which is performed. Summarizing all the performed experiments and the relevant parameters in a table would be helpful.

- The results section is confusing: I miss the connection between Sect 4.1-4.2 (which are quite short) and the rest of the section, which is much more clear and goes into the details of the comparison for the different model configurations and possible reasons for deviations. I would suggest to revise Sect. 4, trying to set a common thread through the whole section.

- The downscaling of dust in EMAC is mentioned in the result section and in the conclusion, but it is not discussed in detail. It seems to be an important issue and shall be discussed in Sect. 3.

- Another interesting point which is mentioned in the conclusions but not discussed in sufficient detail is sea salt composition in the aerosol model and how it can be "tuned"

using satellite data.

SPECIFIC COMMENTS:

There are several sentences which are hard to interpret and more precise statements are sometimes desirable. See detailed suggestions in the following:

P1.L20: this sentence is unclear: a consistent representation of tropospheric and stratospheric aerosol in the model and the good agreement with observations are two different things. You could have a model which represents both domains consistently but compares badly with the observations, and the other way round. I would suggest rephrasing this, stressing that you have a consistent model in terms of aerosol representation (this is a plus) AND that the results reproduce satellite observations well (this is another plus).

P1.L24: you should also explain why radiation is reduced (scattering processes?).

P1.L26: please summarize what are the scope and the goals of this initiative and add a reference, if available.

P1.L31: I would rephrase this as: "like the EMAC model (Bruhl et al., 2015)".

P2.L17: please provide the exact wavelength range.

P2.L33: do you mean that the extinction in cloud-free fraction is attributed to sulfate aerosol? If yes, please rephrase and make it more explicit.

P3.L30: I would mention that AERONET is recognized as the reference dataset for validating satellite products and cite Holben et al. (1998)

P4.L8: please identify the vertical resolutions with a number (L90, L31), that you can refer to in the rest of the paper.

P4.L16: is this the wet or the dry radius?

P4.L18: how are the optical properties calculated? Please provide more details.

P4.L20: given their relevant role in this study, more details on the dust emission parametrization should be given here.

P4.L30: does this generate any inconsistency/discontinuity in the emissions at 200 hPa? Please clarify.

P4.L31: please provide references for the various emission inventories mentioned in this paragraph.

P5.L18: it is not clear what has been downscaled here and why.

P5.L22: please add in which Figure of Bingen et al. this is shown.

P12.L16: which dust size distributions parameter are adopted in the model?

P13.L8: horizontal or vertical resolution? What is the expected outcome of these simulations? Are further publications planned? Please elaborate more on this sentence.

P13.L16: where does this conversion factor come from? It looks like an important issue, but it is mentioned for the first time in the conclusions.

TECHNICAL CORRECTIONS:

P1.L9: "EMAC" acronym is not defined at the first occurrence.

P1.L10: "such as" instead of "like" (you intend inclusion, not comparison).

P1.L13: "the observations".

P2.L6: add "The present paper is organized as follows:" or similar.

Fig.4: red and purple are very hard to distinguish, please consider a different color (or dash pattern).

P12.L24: I would simply write "at T106L31 resolution" and use this notation consistently through the paper.

---

## Referee Comment (RC2) · Anonymous Referee #1 · 19 Jun 2018

The manuscript presents some simulation results for aerosol optical properties obtained with different configurations of the EMAC model and compares them to a variety of satellite products. Clearly, the topic of atmospheric aerosols is appropriate for ACP. My recommendation is, however, to not accept the manuscript for publication in ACP, mainly because its scientific aim is unclear. The introduction of the paper doesn't state an aim, and doesn't formulate a scientific question. It provides some background information in particular on stratospheric aerosols, satellite data, and on earlier studies with EMAC. Furthermore, it makes some general statements, e.g. concerning the usefulness of specific satellite data "to validate and optimize assumptions . . . in the model", of which it is unclear if they are a conclusion of some earlier work or just the opinion

of the authors. No knowledge gaps are mentioned, no strategy how to tackle gaps, and how this work makes progress in comparison to the rich body of literature on the topic. Even the reference to an earlier paper with involvement of several of the authors remains vague and it is unclear what the present study may add ("Some aspects . . . of this study have been addressed in Bingen et al. (2017)"). In section 5 (and also in the abstract) some "conclusions" are actually drawn, but in several cases they do not seem to be backed up thoroughly by the main body of the manuscript. In addition, there are very few statements that could be understood as knowledge gained on the atmosphere. And if they can (like "The total AOD in the visible . . . is very sensitive to aerosol water and the composition of sea salt."), they tend towards being very general and again it is unclear how potential discoveries in this study relate to earlier works. Most of the concluding statements relate to model tuning ("simply assuming a factor of 2 for conversion . . . is too crude"), it is unclear how general or model specific they are, and like in this case they are not well developed in the rest of the paper. Because I understand the manuscript as mostly related to model evaluation and tuning I would suggest the authors consider resubmitting it to a more model development related journal like GMD or JAMES, but even in this case I think large parts of the manuscript would need to be rewritten. In the following I will provide a list of further issues I see with the manuscript:

- Abstract: The abstract contains some statements that can be considered as conclusions, e.g. "sulfate particles from . . . volcanic eruptions dominate the interannual variability of aerosol extinction . . .". But if this is considered important enough to make it to the abstract: why doesn't it appear in the "conclusions"? And is it a new discovery?

- P3L3: "The development work of these CDRs showed . . ." The formulation is odd. How did the work show this importance? And why does it come to this non-linearity in the averaging process? Wouldn't this depend on the way averages are built? Any reference for it?

- P3L25: "size and optical thickness on a 10 km grid". Shouldn't size be provided with

some vertical resolution?

- Section 3: Model Setup. It is not sufficiently clear from this section how observations are used in the simulations. Vague formulations are used in many places, like "use of data . . . for input and validation", "aerosol module parameters . . . were optimized on the basis of satellite data". Which aerosol parameters are actually prognostic quantities, which data are prescribed how (e.g. as boundary or initial conditions?) in the simulation process, which are used how to tune which model parameters. It is important to be very specific here, also to understand how dependent or independent the simulation results are from the data used for evaluation. (see also below)

- P4L6: "we used different model resolution to improve the dust simulations". Sounds odd. I guess one can assume that higher resolution might improve simulations, but this is not what is said, here.

- P4L16: ". . . particle radius of 1.6 um to avoid too fast sedimentation . . ." Again an odd formulation. Would any other parameter lead to too fast sedimentation? And would this parameter be chosen differently based on observed particle size distributions?

- P4L25-35: ". . . superimposed to the simulated SO2 . . ."; "boundary conditions are taken from . . ."; "Marine DMS . . . is also included . . ." Again, these formulations are not specific enough. What does that mean? Are simulated fields simply updated at eruption times? For which boundaries are the observations used? How is marine DMS used? In terms of emissions? Is this important when SO2 concentrations are anyhow newly "superimposed" after every eruption? And what does all this use of observations mean for the evaluation of the simulations?

- P5L5 "due to transport" Which transport is meant, here? From the troposphere to the stratosphere?

- Fig. 1 shows comparisons of extinction profiles for different wavelengths. However, this is not discussed in the text. What do we learn from the different comparisons?

[Figure]

- P5L16 "account for dust in a proper way, also with respect to particle size". What would be a proper way to account for the particle size of dust? And if you say also, what else?

- P5L17 ff: The way the "downscaling" is described here is very misleading. Only much later in the manuscript one learns that the authors have just multiplied the extinction with some factor to obtain a better comparison to observations. It is also not clear where this sensitivity to model resolution comes from. Why does the convection lead to very different transport for a relatively modest change of horizontal resolution? What are the tuning parameters? And how does the tuning of the convection parameterization for fitting sulfur transport influence other important quantities like the radiative balance?

- P5L22, reference to Bingen et al. (2017). It is nice to know that other things are shown elsewhere, but it would be more important to get to know what has been learned elsewhere and what additional knowledge is provided by the figures in this paper.

- P7L3, "our findings that desert dust is also important for the UTLS". A very vague finding. One could try to quantify it. A first step could be to look at difference plots of Fig. S1 (center and bottom panels).

- P7L10, "global radiative forcing" I guess this is probably instantaneous forcing from a double radiation call? It would be important to spell this out.

- P7L13, "Green lines and symbols show … observations like …" There are no symbols in the figure. And what means "observations like"? Be specific. This is supposed to be a scientific paper.

- P7L18, "This is clearly seen in Fig. 4 …" How do I see in Fig. 4 which part of the AOD is transport-related? And how do I see the monsoon effect if only 20S-20N and 45N-70N are shown, not the monsoon region?

- Fig. 3 and 4: Legends would be nice.

- Fig. 4: How is "stratospheric AOD" defined?

- Caption of Fig. 4: All other caption don't include interpretation. And what means "differ mostly"?

- P9L6: "a clear signal from biomass burning organic aerosol. It's not clear to me from just looking at the figure. Please explain why this is clear.

- P11L10ff: There is no motivation provided for the change to resolution T106. How would the problem of tuning convection mentioned for T42 vs. T63 affect T106?

- P13L8, "additional (ongoing) simulations". What do I make of the "ongoing"? Either the simulations are ready to be used for scientific interpretation or not.

---

## Author Comment (AC1) · 31 Jul 2018

**1   Introduction**

The reviewers are right that some sections are too short because the manuscript was written under time pressure. We also understand that the text might be more precise and that several clarifications are needed, and we thank the Referees for their effort to specify what is odd or missing and what causes misunderstandings in our formulation. In the revised version we would follow the suggestions of Referee 2, and also replace preliminary results in the figures by the full simulation noted as 'ongoing' in the

published version criticized be reviewer 1.

Concerning Referee 1's suggestion to submit this manuscript to another journal, our opinion is that GMD is not an adequate one for this study because it addresses both the satellite and modelling community and its focus is on presenting model results and observations, not on model development. This will be addressed more clearly in the revised abstract, introduction, results section and conclusions. Therefore, we believe that the choice of ACP is the right one to reach the concerned communities. One important aspect here is that the model is able to calculate observed quantities at the original wavelengths of the instruments, providing consistent information, also for radiative forcing.

The suggestion for revised abstract and conclusions is:

[revised manuscript text omitted]

**2 Reviewer 1, questions and answers**

**2.1 General comments**

*The introduction of the paper doesn't state an aim, and doesn't formulate a scientific question. It provides some background information in particular on stratospheric aerosols, satellite data, and on earlier studies with EMAC. Furthermore, it makes some general statements, e.g. concerning the usefulness of specific satellite data "to validate and optimize assumptions . . . in the model", of which it is unclear if they are a conclusion of some earlier work or just the opinion of the authors. No knowledge gaps are mentioned, no strategy how to tackle gaps, and how this work makes progress in*

*comparison to the rich body of literature on the topic. Even the reference to an earlier paper with involvement of several of the authors remains vague and it is unclear what the present study may add ("Some aspects . . . of this study have been addressed in Bingen et al. (2017)").* The introduction is expanded and clarified, examples see below in the reply to reviewer 2. Now the third sentence in the introduction is: "The aim of the present paper is to use jointly model simulations and satellite observations, taking into account the multiple spectral channels of the instruments to better understand of the spatio-temporal evolution of the stratospheric aerosol burden and the contribution of the different aerosol types to the observed dynamical aerosol patterns at the different altitudes. Most earlier studies..."

*In section 5 (and also in the abstract) some "conclusions" are actually drawn, but in several cases they do not seem to be backed up thoroughly by the main body of the manuscript. In addition, there are very few statements that could be understood as knowledge gained on the atmosphere. And if they can (like "The total AOD in the visible . . . is very sensitive to aerosol water and the composition of sea salt."), they tend towards being very general and again it is unclear how potential discoveries in this study relate to earlier works.* Text modified for clarity. An important conclusion is that from multiple wavelength observations the enhanced size of wet dust particles simulated by the model in the monsoon region can be detected. Please see also the revised version of abstract and conclusions in the introduction of this comment.

*Most of the concluding statements relate to model tuning ("simply assuming a factor of 2 for conversion . . . is too crude"), it is unclear how general or model specific they are, and like in this case they are not well developed in the rest of the paper.* The factor of 2 refered to a relation of observations at different wavelength often used. The other factor of 0.33 is for correction of model output if a too coarse resolution is selected.

*Because I understand the manuscript as mostly related to model evaluation and tuning I would suggest the authors consider resubmitting it to a more model development related journal like GMD or JAMES.* We don't agree, see second paragraph of introduction above. Text modified to avoid misunderstandings.

2.2   Specific comments

*- Abstract: The abstract contains some statements that can be considered as conclusions, e.g. "sulfate particles from . . . volcanic eruptions dominate the interannual variability of aerosol extinction . . .". But if this is considered important enough to make it to the abstract: why doesn't it appear in the "conclusions"? And is it a new discovery?* Abstract and conclusions rewritten for clarity, see above in introduction of this comment.

*- P3L3: "The development work of these CDRs showed . . ." The formulation is odd. How did the work show this importance? And why does it come to this non-linearity in the averaging process? Wouldn't this depend on the way averages are built? Any reference for it?* We revised the formulation of the discussion taking into account the questions posed by the Referee, and rearranged somewhat the text. We are not sure about what the Referee means by "nonlinearity", since all is mainly about how the observed patterns are visible in small bins or lost when the averaging occurs in a large volume compared to the size of the pattern (e.g.: a volcanic plume), which is just a linear problem, but we hope that our revision clarifies the text and removes all kind of misunderstanding. We also added a reference to the paper describing the whole study, in order to avoid duplication with this previous paper (Bingen et al., 2017).

*- P3L25: "size and optical thickness on a 10 km grid". Shouldn't size be provided with some vertical resolution?* Both optical thickness and size are retrieved as vertical column values. Size is not resolved vertically, but is represented by fraction of fine and coarse mode aerosol in the total.

*- Section 3: Model Setup. It is not sufficiently clear from this section how observations are used in the simulations. Vague formulations are used in many places, like "use of*

*data . . . for input an" validation", "aerosol module parameters . . . were optimized on the basis of satellite data". Which aerosol parameters are actually prognostic quantities, which data are prescribed how (e.g. as boundary or initial conditions?) in the simulation process, which are used how to tune which model parameters. It is important to be very specific here, also to understand how dependent or independent the simulation results are from the data used for evaluation. (see also below)* The text is improved and expanded a lot, providing more details and references, examples see below.

*- P4L6: "we used different model resolution to improve the dust simulations". Sounds odd. I guess one can assume that higher resolution might improve simulations, but this is not what is said, here.* The text is modified here. This points to a problem that is often ignored by modellers. New text: "In contrast to Jöckel et al. (2016), who use stratospheric aerosol extinction climatologies derived from observations, in our model setup aerosol and its optical properties are calculated from precursor gases and emissions. As dust reaching the upper troposphere/lower stratosphere region (UTLS) turned out to be sensitive to model resolution, we used different model resolutions: the T42 resolution (spectral, $2.75^o$ in latitude and longitude) of the previous studies, T63 resolution ($1.88^o$), the standard resolution for the stratosphere used in this study and T106 resolution ($1.1^o$) for a one year sensitivity test. The vertical grid has 90 layers from the surface up to 0.01 hPa ( 80 km altitude, short L90) with finest resolution in the boundary layer and near the tropopause. For T106 only simulations with the low top model version with 31 levels up to 30 km altitude (L31), the setup used by Klingmüller et al. (2018) which is well tested regarding the representation of tropospheric aerosol are discussed here in detail."

*- P4L16: ". . . particle radius of 1.6 um to avoid too fast sedimentation . . ." Again an odd formulation. Would any other parameter lead to too fast sedimentation? And would this parameter be chosen differently based on observed particle size distributions?* Text clarified: "The boundary between accumulation mode and coarse mode, a model

parameter, is set at a dry particle radius of 1.6 $\mu$m to avoid too fast sedimentation of a too large coarse mode fraction in case of major volcanic eruptions. For dust sensitivity studies in T106 which focus on the troposphere, also a boundary of 1.0 $\mu$m is used. The mode parameters are used for every aerosol type and listed for convenience in Table S1 of the supplement. "

*- P4L25-35: ". . . superimposed to the simulated SO2 . . ."; "boundary conditions are taken from . . ."; "Marine DMS . . . is also included . . ." Again, these formulations are not specific enough. What does that mean? Are simulated fields simply updated at eruption times? For which boundaries are the observations used? How is marine DMS used? In terms of emissions? Is this important when SO2 concentrations are anyhow newly "superimposed" after every eruption? And what does all this use of observations mean for the evaluation of the simulations?* This paragraph is expanded and corrected to be unique. "superimposed" is replaced by "added as volume mixing ratio". We provide references for the data and mention uncertainties from the model resolution. $SO_2$ from DMS is important in volcanicly quiet periods, i.e. when gaps between the eruptions exceeds about a month. For DMS we add in the text: "... model, using a module for exchange fluxes between seawater and atmosphere by Pozzer et al. (2006) and the Lana et al (2011) climatology".

*- P5L5 "due to transport" Which transport is meant, here? From the troposphere to the stratosphere?* Section expanded, "convective" and "gradual uplift to UTLS" added.

*- Fig. 1 shows comparisons of extinction profiles for different wavelengths. However, this is not discussed in the text. What do we learn from the different comparisons?* This part is expanded, providing more details on GOMOS.

*- P5L16 "account for dust in a proper way, also with respect to particle size". What would be a proper way to account for the particle size of dust? And if you say also, what else?* This section is rewritten, including new frames for Fig.2 which demonstrate the connection between bigger wet dust particles in the monsoon region and a relatively

larger extinction in near IR compared to the visible range. Figure 2 contains now also the contribution of wet dust and wet sulfate at 2 wavelengths individually, together with the observations and the calculated median wet radius in the accumulation mode.

*- P5L17 ff: The way the "downscaling" is described here is very misleading. Only much later in the manuscript one learns that the authors have just multiplied the extinction with some factor to obtain a better comparison to observations. It is also not clear where this sensitivity to model resolution comes from. Why does the convection lead to very different transport for a relatively modest change of horizontal resolution? What are the tuning parameters? And how does the tuning of the convection parameterization for fitting sulfur transport influence other important quantities like the radiative balance?* This is addressed in sections 3, 4.2, conclusions and the supplement now. A reference to the used Tiedke scheme is given. We don't tune the convection scheme, we just modify the output of the routine providing SAOD in case of the coarse resolution.

*- P5L22, reference to Bingen et al. (2017). It is nice to know that other things are shown elsewhere, but it would be more important to get to know what has been learned elsewhere and what additional knowledge is provided by the figures in this paper.* This was misplaced here. There is a reference to earlier work in section 4.1 now. The novel results are presented in the new Figure 2 where the effect of big wet dust particles can be identified in observed and simulated extinction in the monsoon area.

*- P7L3, "our findings that desert dust is also important for the UTLS". A very vague finding. One could try to quantify it. A first step could be to look at difference plots of Fig. S1 (center and bottom panels).* See above.

*- P7L10, "global radiative forcing" I guess this is probably instantaneous forcing from a double radiation call? It would be important to spell this out.* This is mentioned in section 3 now.

*- P7L13, "Green lines and symbols show . . . observations like . . ." There are*

*no symbols in the figure. And what means "observations like"? Be specific. This is supposed to be a scientific paper.* We have increased the size of the symbols for the annual averages. The curves and symbols for observations are taken from the references provided in the text ("like" removed).

*- P7L18, "This is clearly seen in Fig. 4 . . ." How do I see in Fig. 4 which part of the AOD is transport-related? And how do I see the monsoon effect if only 20S-20N and 45N-70N are shown, not the monsoon region?* Figure 4 is expanded, thanks for this important advice. To achieve this we had to process the OSIRIS data directly (and to include a coauthor) in addition to GOMOS which has not enough coverage for narrow latitude bands. With the new data the monsoon effect is clearly visible, including the problems concerning the overestimated dust with low T42 resolution of the older studies.

*- Fig. 3 and 4: Legends would be nice.* We improved the color scheme and the description. Unfortunately our graphics package cannot create legends in a convenient way, but if the reviewer or the editor insists on it, we have a postprocessing tool.

*- Fig. 4: How is "stratospheric AOD" defined?* Lower boundaries of integrals now provided in the text, dependent on latitude (sorry, somehow this was lost in the manuscript). Upper boundary is at the top of the Junge layer, i.e. about 30km.

*- Caption of Fig. 4: All other caption don't include interpretation. And what means "differ mostly"?* This is obsolete now.

*- P9L6: "a clear signal from biomass burning organic aerosol". It's not clear to me from just looking at the figure. Please explain why this is clear.* In the figure are spikes in the lower stratosphere in September, correlated with maxima in the mixing ratios of absorbing organic aerosol. A figure on that will be in the supplement.

*- P11L10ff: There is no motivation provided for the change to resolution T106. How would the problem of tuning convection mentioned for T42 vs. T63 affect T106?* We

don't tune convection. Resolution T106L31 was the standard setup for tropospheric aerosol modelling in earlier studies (indicated now in section 3). We like to show here that this setup and the stratospheric setup T63L90 are consistent concerning the comparison with IASI and ATSR.

*- P13L8, "additional (ongoing) simulations". What do I make of the "ongoing"? Either the simulations are ready to be used for scientific interpretation or not.* Removed. The results in the figures are replaced by the ones of the completed "ongoing" simulation, see also introduction of this comment.

**3  Reviewer 2, questions and answers**

**3.1  General comments**

*- The introduction is too short and does not put this study in the context of existing literature. Similar papers on the subject shall be cited and related to the work presented in the manuscript.* The introduction is expanded and misleading words (e.g. in line 6, page 2) were replaced. In line 23, page 1 two more references are cited, in line 26 the following is inserted: "Most earlier studies focus on the effects of major volcanic eruptions like Pinatubo (e.g. Aquila et al., 2014, English et al., 2013). For the post Pinatubo-period with only medium size eruptions Mills et al (2016, 2017) present simulations with the chemistry climate model WACCM (Whole Atmosphere Community Model) with interactive aerosol using estimates for volcanic injections mostly from nadir sounders. That and the present ....".

*- It is also not immediately clear what are the novelty aspect of this study and what is its added value to the current knowledge in the field. It should be stated that the paper focuses on the evaluation of the EMAC model in different configurations, specifically on the aspects controlling the optical and radiative properties of stratospheric aerosol.*

Please see revised abstract and conclusions in the introduction of this comment. Model evaluation is only one aspect. Novel is e.g. the clear indication of the presence of dust in the LS from multiwavelength observations and model simulations (abstract and conclusions).

*- The description of model setup is also not very exhaustive: it is not clear for example how the SO2 plumes from the data are used in the model (last paragraph of Sect. 3). It is also not mentioned which time period is covered by the simulations (although this can be found out later in the figures) and whether a nudging technique has been used.* Concerning nudging, we include in line 10 of section 3: "In all simulations, except the T42L90 one of the previous studies, the meteorology below about the 100 hPa level is nudged to the reanalysis ERA-Interim." The part on $SO_2$ injections is considerably expanded providing references (see reply to reviewer 1).

*- As far as I could understand, the setup has been derived from one of the simulations in Jöckel et al. (2016). If this is the case, I would recommend to write that more explicitly at the beginning of Sect. 3. The rest of the section could then discuss just the differences and the additional features considered for the present study. The choice of particular configuration settings shall also be motivated in view of the analysis which is performed. Summarizing all the performed experiments and the relevant parameters in a table would be helpful.* The most important difference to Jöckel et al. (2016) is now in second sentence of section 3: "In contrast to Jöckel et al. (2016), who use stratospheric aerosol extinction climatologies derived from observations, in our model setup aerosol and its optical properties are calculated from precursor gases and emissions." The definitions and names of the simulations are clarified.

*- The results section is confusing: I miss the connection between Sect 4.1-4.2 (which are quite short) and the rest of the section, which is much more clear and goes into the details of the comparison for the different model configurations and possible reasons for deviations. I would suggest to revise Sect. 4, trying to set a common thread through the whole section.* Sections 4.1 and 4.2 are expanded and related to section 4.3 with

the most important results. Section 4.3 contains now additional satellite data to support the findings that dust transported by the monsoon can be important.

*- The downscaling of dust in EMAC is mentioned in the result section and in the conclusion, but it is not discussed in detail. It seems to be an important issue and shall be discussed in Sect. 3.* The resolution dependent scaling is mentioned in detail for the volcanic $SO_2$ sink, an additional problem we detected after submission of the first version of the manuscript, in section 3. In section 4.2 and the supplement details scaling of stratospheric dust extinction for low resolution are given.

*- Another interesting point which is mentioned in the conclusions but not discussed in sufficient detail is sea salt composition in the aerosol model and how it can be "tuned" using satellite data.* We now provide text and references in section 3 and a table in the supplement.

3.2   Specific comments

*There are several sentences which are hard to interpret and more precise statements are sometimes desirable. See detailed suggestions in the following:*

*P1.L20: this sentence is unclear: a consistent representation of tropospheric and stratospheric aerosol in the model and the good agreement with observations are two different things. You could have a model which represents both domains consistently but compares badly with the observations, and the other way round. I would suggest rephrasing this, stressing that you have a consistent model in terms of aerosol representation (this is a plus) AND that the results reproduce satellite observations well (this is another plus).* Sentence changed, see revised abstract in introduction of this comment.

*P1.L24: you should also explain why radiation is reduced (scattering processes?).* Done.

*P1.L26: please summarize what are the scope and the goals of this initiative and add a reference, if available.* Timmreck et al. (2018) and Kremser et al. (2016) references given.

*P1.L31: I would rephrase this as: "like the EMAC model (Brühl et al., 2015)".* Done.

*P2.L17: please provide the exact wavelength range.* Done.

*P2.L33: do you mean that the extinction in cloud-free fraction is attributed to sulfate aerosol? If yes, please rephrase and make it more explicit.* We thank the referee for this pertinent question. We clarify earlier in the text the way to retrieve the extinction and changed the expression "total aerosol extinction" in the more adequate expression "total extinction from non-gaseous species". We also explain now how the type of particulate matter is inferred, which makes possible the derivation of separate CDRs for the total extinction, the aerosol fraction, and the polar stratospheric cloud fraction. It should be noted that the extinction by particulate matter is retrieved using a parameterization that doesn't require any knowledge of the aerosol type. So, the statement concerning the sulfate aerosols simply refers to the well-known fact that sulfate aerosols is the most common aerosol type observed in the stratosphere, but we do not claim bringing any new information on the aerosol type. Hence the use of the qualification "in good approximation": at this stage, we cannot infer more precisely the nature of the aerosol particles. A warning is also added about the limitations of the criteria used for the cloud detection, and the risk for cloud contamination in the aerosol CDR. Please note that all these aspects are detailed in the different references cited in the text, so that we don't think that more detail is needed on the way the retrieval algorithm is implemented.

*P3.L30: I would mention that AERONET is recognized as the reference dataset for validating satellite products and cite Holben et al. (1998)* Done.

*P4.L8: please identify the vertical resolutions with a number (L90, L31), that you can refer to in the rest of the paper.* Done.

[Figure]

*P4.L16: is this the wet or the dry radius?* "dry" added here in text.

*P4.L18: how are the optical properties calculated? Please provide more details.* Done, added in text: "...calculated using Mie-theory-based lookup tables consistent with the selected size distribution widths of the modes. The resulting optical depths, single scattering albedos and asymmetry-factors are used in radiative transfer calculations...."

*P4.L20: given their relevant role in this study, more details on the dust emission parametrization should be given here.* We have expanded the text to "The mineral dust emissions are calculated online using the emission scheme of Astitha et al. (2012) which builds on previous studies by Pérez et al. (2006), Spyrou et al. (2010), Laurent et al. (2008, 2010), Marticorena et al. (1997), Zender et al. (2003) and Tegen (2002). The emission scheme parameterizes saltation bombardment and aggregate disintegration by sand blasting combining the surface friction velocity with descriptions of land cover type, clay fraction of the soil and vegetation cover. For an improved representation of dust at higher resolution, we adopted the updates presented by Klingmüller et al. (2018) in the T106 L31 simulation."

*P4.L30: does this generate any inconsistency/discontinuity in the emissions at 200 hPa? Please clarify.* This effect is secondary and in most cases smeared out by transport. I have added "monthly" in the text. The Diehl climatology is mostly for volcanoes outgassing over long periods, for explosive events it introduces local artifacs in the upper troposphere which have negligible effects in the LS.

*P4.L31: please provide references for the various emission inventories mentioned in this paragraph.* The detailed description can be found in Jöckel et al 2016. This covers several pages including references and is not the main focus of this study. "CCMI" is added in the text.

*P5.L18: it is not clear what has been downscaled here and why.* This section is rewritten using the results of the T63L90 simulation. Results for T42L90, the resolution of earlier studies, are shown in the supplement and mentioned at the end of the section.

*P5.L22: please add in which Figure of Bingen et al. this is shown.* Done at end of
section 4.1. In section 4.2 this is skipped.

*P12.L16: which dust size distributions parameter are adopted in the model?* The dust
size distribution is calculated online resulting from the physical and chemical processes
acting on the aerosol modes. Fixed parameters are the widths of the log-normal modes
and the dry radii separating the modes. The size distribution can be adjusted by mod-
ifying these parameters, but also by modifying parameters of relevant processes such
as emission, deposition, coagulation and hygroscopic growth. For clarification, we have
added the following text to line 18 on page 12: "This could involve modifying the param-
eters of the log-normal modes, i.e., their widths and boundaries, but also reassessing
the parametrisation of relevant processes such as emission, deposition, coagulation
and hygroscopic growth, or even adding an extra mode for extremely coarse particles
which can be relevant close to dust sources." Furthermore, we refer to a table in the
supplement with the mode parameters in section 3.

*P13.L8: horizontal or vertical resolution? What is the expected outcome of these sim-
ulations? Are further publications planned? Please elaborate more on this sentence.*
This sentence was premature. I had estimated the typical results for the whole time-
series from an ongoing incomplete simulation.

*P13.L16: where does this conversion factor come from? It looks like an important
issue, but it is mentioned for the first time in the conclusions.* This issue is now ad-
dressed also in section 4.4. The factor is often used in the AEROCOM community but
not physically based.

3.3  Technical corrections:

*P1.L9: "EMAC" acronym is not defined at the first occurrence.* In the abstract this
should be considered as a name because the full expression of this acronym of

acronyms given in section 3 is too long (3 lines).

*P1.L10: "such as" instead of "like" (you intend inclusion, not comparison).* Done.

*P1.L13: "the observations".* Rewritten.

*P2.L6: add "The present paper is organized as follows:" or similar.* Done, and misleading words replaced.

*Fig.4: red and purple are very hard to distinguish, please consider a different color (or dash pattern).* Color scheme now as in Fig. 3.

*P12.L24: I would simply write "at T106L31 resolution" and use this notation consistently through the paper.* Done.

**4 Additional references**

Aquila, V., Oman, L. D., Stolarski, R. S., Colarco, P. R., and Newman, P. A.: Dispersion of the volcanic sulfate cloud from a Mount Pinatubolike eruption. J. Geophys. Res., 117, D06216. https://doi.org/10.1029/2011JD016968, 2012

English, J. M., Toon, O. B., and Mills, M. J.: Microphysical simulations of large volcanic eruptions: Pinatubo and Toba. Journal of Geophys. Res. Atmos, 118, 1880–1895. https://doi.org/10.1002/jgrd.50196, 2013

Lana, A., Bell, T. G., Simó, R., Vallina, S. M., Ballabrera-Poy, J., Kettle, A. J., Dachs, J., Bopp, L., Saltzman, E. S., Stefels, J., Johnson, J. E., and Liss, P. S.: An updated climatology of surface dimethlysulfide concentrations and emission fluxes in the global ocean, Global Biogeochem. Cy., 25, GB1004, doi:10.1029/2010GB003850, 2011.

Laurent, B., Marticorena, B., Bergametti, G., Léon, J. F., and Mahowald, N. M.: Modeling mineral dust emissions from the Sahara desert using new surface properties and soil database, J. Geophys. Res.-Atmos., 113, d14218, https://doi.org/10.1029/2007JD009484, 2008.

Laurent, B., Tegen, I., Heinold, B., Schepanski, K., Weinzierl, B., and Esselborn, M.: A model study of Saharan dust emissions and distributions during the SAMUM-1 campaign, J. Geophys. Res.- Atmos., 115, d21210, https://doi.org/10.1029/2009JD012995, 2010.

Marticorena, B., Bergametti, G., Aumont, B., Callot, Y., N'Doumé, C., and Legrand, M.: Modeling the atmospheric dust cycle: 2. Simulation of Saharan dust sources, J. Geophys. Res.-Atmos., 102, 4387–4404, https://doi.org/10.1029/96JD02964, 1997.

Mills, M.J., A. Schmidt, R. Easter, S. Solomon, D. E. Kinnison, S. J. Ghan, R. R. Neely III, D. R. Marsh, A. Conley, C. G. Bardeen, and A. Gettelman: Global volcanic aerosol properties derived from emissions, 1990–2014, using CESM1(WACCM) , J. Geophys. Res. Atmos., 121, 2332–2348, doi:10.1002/2015JD024290, 2016.

Mills, M. J., Richter, J. H., Tilmes, S., Kravitz, B., MacMartin, D. G., Glanville, A. A., Kinnison, D. E.:. Radiative and chemical response to interactive stratospheric sulfate aerosols in fully coupled CESM1(WACCM). J. Geophys. Res. Atmos., 122, 13,061– 13,078. https://doi.org/10.1002/2017JD027006, 2017

Pérez, C., Nickovic, S., Baldasano, J. M., Sicard, M., Rocadenbosch, F., and Cachorro, V. E.: A long Saharan dust event over the western Mediterranean: Lidar, Sun photometer observations, and regional dust modeling, J. Geophys. Res.-Atmos., 111, d15214, https://doi.org/10.1029/2005JD006579, 2006.

Pozzer, A., Jöckel, P., Sander, R., Williams, J., Ganzeveld, L., and Lelieveld, J.: Technical Note: The MESSy-submodel AIRSEA calculating the air-sea exchange of chemical species, Atmos. Chem. Phys., 6, 5435–5444, doi:10.5194/acp-6-5435-2006, 2006

Ridley, D. A., et al., Total volcanic stratospheric aerosol optical depths and implications for global climate change, Geophys. Res. Lett., 41, 7763–7769,

doi:10.1002/2014GL061541, 2014

Spyrou, C., Mitsakou, C., Kallos, G., Louka, P., and Vlastou, G.: An improved limited area model for describing the dust cycle in the atmosphere, J. Geophys. Res.-Atmos., 115, d17211, https://doi.org/10.1029/2009JD013682, 2010.

Tegen, I.: Impact of vegetation and preferential source areas on global dust aerosol: Results from a model study, J. Geophys. Res., 107, 4576, https://doi.org/10.1029/2001JD000963, 2002.

Zender, C. S., Bian, H., and Newman, D.: Mineral Dust Entrainment and Deposition (DEAD) model: Description and 1990s dust climatology, J. Geophys. Res. Atmos., 108, 4416, https://doi.org/10.1029/2002JD002775, 2003.